# Improving the Performance of Polymer Solar Cells with Benzo[*ghi*]perylenetriimide-Based Small-Molecules as Interfacial Layers

**DOI:** 10.3390/polym14204466

**Published:** 2022-10-21

**Authors:** Yang-Yen Yu, Hung-Cheng Chen, Kai-Yu Shih, Yan-Cheng Peng, Bing-Huang Jiang, Chao-I Liu, Ming-Wei Hsu, Chi-Ching Kuo, Chih-Ping Chen

**Affiliations:** 1Department of Materials Engineering, Ming Chi University of Technology, New Taipei City 24301, Taiwan; 2Department of Applied Chemistry, National University of Kaohsiung, Kaohsiung 81148, Taiwan; 3Cagu International Co., Ltd., Kaohsiung 80652, Taiwan; 4Research and Development Center of Smart Textile Technology, Institute of Organic and Polymeric Materials, National Taipei University of Technology, Taipei 10608, Taiwan

**Keywords:** benzo[ghi]perylenetriimide, interface modification layer, polymer solar cells, green energy

## Abstract

In this study, we prepared three benzo[*ghi*]perylenetriimide (BPTI) conjugated molecules as electron-transporting surface-modifying layers for polymer solar cells (PSCs). These three BPTI derivatives differed in the nature of their terminal functionalities, featuring butylamine (C_3_NH_2_), propylammonium iodide (C_3_NH_3_I), and butyldimethylamine (C_3_DMA) units, respectively. We evaluated the optoelectronic properties of PTB7-Th: PC_71_BM blends modified with these interfacial layers, as well as the performance of resulting PSCs. We used UV–Vis spectroscopy, atomic force microscopy, surface energy analysis, ultraviolet photoelectron spectroscopy, and photoelectric flow measurements to examine the phenomena behind the changes in the optoelectronic behavior of these blend films. The presence of a BPTI derivative changed the energy band alignment at the ZnO–active layer interface, leading to the ZnO film behaving more efficiently as an electron-extraction electrode. Modifying the ZnO surface with the BPTI-C_3_NH_3_I derivative resulted in a best power conversion efficiency (PCE) of 10.2 ± 0.53% for the PTB7-Th:PC_71_BM PSC (cf. PCE of the control device of 9.1 ± 0.13%). In addition, modification of a PM6:Y6:PCBM PSC with the BPTI-C_3_NH_3_I derivative increased its PCE from 15.6 ± 0.25% to 16.5 ± 0.18%. Thus, BPTI derivatives appear to have potential as IFLs when developing high-performance PSCs, and might also be applicable in other optoelectronic devices.

## 1. Introduction

Because of their high flexibility, high stretchability, excellent mechanical properties, light weight, low cost, and amenability to large-area roll-to-roll fast solution manufacturing, polymer solar cells (PSCs) have attracted a great deal of attention in the past decade [1,2,3,4]. Because PSCs can be constructed on flexible and stretchable substrates, they can be integrated with curved subjects for the preparation of wearable electronics. The rapid progress of PSCs has relied mostly on the development of suitable conjugated polymers and small molecules [5,6,7,8]. Controlling the nano-phase segregation of their bulk-heterojunction (BHJ) morphologies and the molecular packing of their blend films has also led to optimization of the power conversion efficiencies (PCEs) of PSCs to greater than 19% [9,10,11,12,13,14,15,16]. Typically, a PSC is constructed with layers of an electrode, an electron transporting layer (ETL), an active layer, a hole transporting layer (HTL), and an electrode. Simply embedding an additional layer between the interfaces of the active layer can alter the morphology of the blend film and the efficiency of its carriers’ extraction, thereby further improving the performance of such PSCs [17,18,19,20,21,22,23,24]. Poly [4,8-bis(5-(2-ethylhexyl)thien-2-yl)benzo [1,2-*b*;4,5-*b*′]dithiophene-2,6-diyl–*alt*–(4-(2-ethylhexyl)-3-fluorothieno [3,4-*b*]thiophene)-2-carboxylate-2-6-diyl] (PTB7-Th or PCE10) and [6,6]-phenyl-C_71_-butyric acid methyl ester (PC_71_BM) are currently the most highly developed materials used as PSC active layers. Incorporating an interface modification layer (IFL) to improve the carrier extraction efficiency in PSCs is a method that is readily applicable to most systems and suitable for fast-manufacturing processing. An excess or unbalanced amount of holes or electrons accumulating in the active layer will significantly affect the short-circuit current density (*J*_SC_) and fill factor (FF) of the corresponding device. To enhance the extraction of such carriers, embedding an IFL can help to decrease the degree of interfacial charge recombination, as well as the interfacial resistance, surface roughness, and number of surface defects. Previous studies have revealed great success when employing perylene-3,4,9,10-tetracarboxylic acid diimides (PDIs) and their derivatives in PSC applications [25,26,27,28]. These materials have strong electron-accepting properties and facilitate efficient charge transport. The use of benzo[*ghi*]perylenetriimide (BPTI) as the IFL of a perovskite solar cell attracted our attention for its similar application in PSCs [28].

Accordingly, in this study, we prepared BPTI derivatives presenting various functional groups—butyldimethylamine (C_3_DMA), butylamine (C_3_NH_2_), and propylammonium iodide (C_3_NH_3_I)—and tested them as IFLs within PSCs (Figure 1). We embedded these BPTI derivatives between the ZnO layer and the active layer. We found that the BPTI derivatives modified the surface properties of the ETL (i.e., ZnO) and altered the growth of the active layer. The N atoms of the BPTI derivatives formed hydrogen bonds with the materials in the active layer, thereby facilitating electron transport and inhibiting carrier recombination at the interface. The I^–^ ion of the C_3_NH_3_I unit appeared to have the effect of eliminating charge accumulation (i.e., hole blocking) and promoting the PSC performance. The presence of the C_3_DMA units decreased the hydrophilicity of the ZnO layer, thereby altering the surface energy of the substrate and changing the blend morphology. We studied the effects of these IFLs on the morphologies and optoelectronic properties of PTB7-Th/PC_71_BM and PM6:Y6:PC_71_BM active layers, as well as the performance of PSCs incorporating them. We observed an enhancement in performance for the PSC containing embedded BPTI-C_3_NH_3_I. The best PSC performance of the BPTI-C_3_NH_3_I–modified PM6:Y6:PCBM featured a PCE of 16.5 ± 0.18%; in comparison, the corresponding control device provided a PCE of 15.6 ± 0.25%. Our results suggest that judicious selection of the IFL can be used to optimize the optoelectronic properties of PSCs, providing a potential pathway for further increases in performance.

## 2. Results and Discussion

We synthesized the three BPTI derivatives according to previously reported procedures [29]. Details of the synthesis and characterization of the BPTI derivatives are available in the Supporting Information. Appendix A present the ^1^H and ^13^C NMR spectra of the BPTI derivatives, supporting their successful synthesis. Appendix A displays the MALDI-TOF mass spectra of BPTI-C_3_NH_2_ and BPTI-C_3_DMA, confirming their purity. Figure 2 provides the UV–Vis absorption spectra of the BPTI derivatives and the UV–Vis transmittance spectra of indium tin oxide (ITO)/ZnO/BPTI substrates. Appendix A provides the UV–Vis absorption spectra of the BPTI derivatives as solutions in CHCl_3_ and in the form of glass/BPTI substrates. As indicated in Figure 2a, the absorptions of the BPTI films were located mainly in the UV region and at wavelengths between 400 and 520 nm. The absorptions of the solid films were slightly red shifted when compared with those in solution status (Appendix A). The absorptions of the thin films were similar on the different substrates, suggesting that the nature of the substrate had only a minimal effect. The absorption spectra of these three BPTI derivatives were similar, and consistent with that of the parent BPTI. Changing the end functionality affected the solubility of the BPTI derivatives and led to changes in their absorption intensities. Next, we deposited the ZnO layer onto the ITO substrate through sol–gel processing of a solution of Zn(OAc)_2_ in 2-methoxyethanol. The resulting ZnO film (thickness: ca. 40 nm) was annealed at 170 °C for 20 min in air prior to deposition of layers of the BPTI derivatives. Solutions of the BPTI derivatives were prepared in CHCl_3_ (CF) at the optimized concentration (0.5 mg mL^−1^***)*** These solutions were deposited on top of the ZnO layers through spin-coating (2000 rpm) in air and then solvent-annealed with CF to induce alignment through self-assembly. The resulting samples were dried (100 °C, 5 min) in a glove box prior to the deposition of the active layer. Figure 2b presents the transmittance (T%) of these samples. The embedding of the BPTI derivatives decreased the values of T% of samples slightly at wavelengths in the region from 400 to 520 nm, which was consistent with absorption of the BPTI derivatives.

We used atomic force microscopy (AFM) and contact angle measurements to study the surficial properties of the ITO/ZnO samples in the absence and presence of the BPTI derivatives. Appendix A displays their tapping-mode AFM images. The root mean square (RMS) surface roughnesses of the unmodified ZnO film and those modified with BPTI-C_3_NH_2_, BPTI-C_3_NH_3_I, and BPTI-C_3_DMA were 14.4, 11.2, 9.4, and 11.3 nm, respectively. Thus, the BPTI-modified ZnO films had the smoother surfaces. A smooth interface between an active layer and an ETL can be beneficial to electron extraction; previous reports have revealed that the hydrophobicity or hydrophilicity of a substrate can greatly affect the blend morphology and device performance [30,31,32]. We performed the contact angle measurements using distilled H_2_O and diiodomethane (CH_2_I_2_, DIM) as probe liquids. We employed the Wu model to calculate the surface energies (*γ*_total_) of the ZnO surfaces and, thereby, investigate the effects of the BPTI derivatives. The value of *γ*_total_ is equal to the sum of the dispersive (*γ*_dispersive_) and polar (*γ*_polar_) components, which we could determine [33]. Table 1 reveals that the water contact angles (*θ*_water_) of the ZnO, ZnO/BPTI-C_3_NH_2_, ZnO/BPTI-C_3_NH_3_I, and ZnO/BPTI-C_3_DMA samples were 40.45, 54.65, 60.60, and 46.76°, respectively. Thus, the values of *θ*_water_ increased after modifying the ZnO film with the BPTI derivative, implying an increase in the hydrophobicity of the respective surface. The values *θ*_DIM_ of the ZnO, ZnO/BPTI-C_3_NH_2_, ZnO/BPTI-C_3_NH_3_I, and ZnO/BPTI-C_3_DMA samples were 26.91, 24.67, 26.95, and 20.74°, respectively. Thus, the presence of BPTI-C_3_DMA significantly increased the lipophilicity of the substrate, due to the presence of the DMA structure on the side chain. The surface energies (*γ*_total_) of the ZnO, ZnO/BPTI-C_3_NH_2_, ZnO/BPTI-C_3_NH_3_I, and ZnO/BPTI-C_3_DMA samples were 71.77, 65.48, 61.93, and 70.32 mN m^−1^, respectively. To double-check the data, we used the Owens–Wendt–Rabel–Kaelble (OWRK) model to calculate the surface energies of the various samples. Table 1 reveals that the results were similar, with the same trends in the changes in the surface energies of the respective samples. The orientation of a small molecule–based IFL can be adjusted through solvent vapor annealing (SA), with the optimized surface properties of the IFL directly affecting the performance of corresponding devices [34]. Because of the similar chemical structures of the three IFLs, we evaluated the effect only of MeOH (polar protic solvent) on BPTI-C_3_NH_3_I, which has high polarity due to its ammonium iodide functionality. Appendix A presents the contact angles and surface energies of the IFLs prepared with and without SA. We observed a higher value of *θ*_water_ and a lower value of *θ*_DIM_ for the sample after SA with CF, suggesting that the ammonium iodide groups were embedded at the bottom of the film. After treatment with MeOH, the sample had a lower value of *θ*_water_ with a higher value of *θ*_DIM_, implying that the ammonium iodide units were distributed mainly on the surface of the IFL. These variations in surface orientation led to different surface energies for the CF- and MeOH-treated samples (60.73 and 62.91 mN m^−1^, respectively). These changes in the contact angles and surface energies would affect the wetting properties of solutions of the active layers. The miscibility of donor and acceptor moieties, a characteristic that can be evaluated from the surface energy, is a major factor that can affect the blend morphology [35,36,37]. Previous reports have indicated that the surface energy of a substrate can alter the phase separation morphology of the active layer [33], with the driving force possibly being a large difference in surface energy (or miscibility) between the two components [38]. Thus, we examined the effect of changes in the surface energies of the substrates upon the variations in their blend film morphologies.

We employed devices with the structure ITO/ZnO/IFL/PTB7-Th:PC_71_BM/MoO_3_/Ag [Figure 3a] to obtain *J*–*V* curves, using an AM 1.5G solar simulator (Peccll PEC-L11, Yokohama, Japan) operated at an illuminating power of 100 mW cm^−2^. Figure 3b and Table 2 summarize the performance data. The control ZnO PSC provided a PCE of 9.1 ± 0.13%, comparable with those of PTB7-Th devices reported previously in the literature. First, we determined the performance of the BPTI-containing devices that had not been subjected to post-solvent treatment. The PCEs of these devices increased slightly relative to that of the control device. After CF-treatment, the PCEs of the devices incorporating BPTI-C_3_NH_2_, BPTI-C_3_NH_3_I, and BPTI-C_3_DMA all increased significantly, reaching 9.6 ± 0.31%, 9.9 ± 0.11%, and 9.3 ± 0.26%, respectively. The improved performance of these devices, relative to the control PSC, was due to significant increases in their FFs. Because of the polarity of the ammonium iodide unit of BPTI-C_3_NH_3_I, we also treated its sample with methanol (MeOH), a solvent of higher polarity, leading to a further improvement in performance, with the best BPTI-C_3_NH_3_I–containing PSC device providing a PCE of 10.2 ± 0.53%. The improvements in the values of *J*_SC_ and FF confirmed that these IFLs had the effect of passivating ZnO defects [39,40]. Figure 3c displays the external quantum efficiency (EQE) spectra of the PSCs incorporating ZnO and BPTI-C_3_NH_3_I-modified ZnO; the photoresponses were consistent with those of the PTB7-Th-based PSCs. From the EQE spectra and the solar flux, we calculated the values of EQE–*J*_SC_ of the PSCs incorporating ZnO and the BPTI-C_3_NH_3_I-modified ZnO to be 15.2 and 15.6 mA cm^−2^, respectively. The mismatch arose from various factors, including the measurement conditions at the solar simulator not being the same as those during the EQE measurements [41]. We performed time-resolved photoluminescence (TRPL) measurements to calculate the effective carrier lifetimes in the blend films and, thereby, determine the phenomena governing their performance. Appendix A displays the TRPL spectra; Appendix A summarizes the respective parameters. The values of *τ*_avg_ of the blend films on the unmodified and BPTI-C_3_NH_2_-, BPTI-C_3_NH_3_I-, and BPTI-C_3_DMA-modified ZnO were 1.202, 1.743, 1.997, and 1.648 ns, respectively. Thus, the carrier lifetimes in the blends increased in the presence of the BPTI derivatives. This finding confirmed that carrier transport was improved when incorporating these IF layers.

To understand why embedding the BPTI derivatives significantly improved the FFs of the devices, we employed ultraviolet photoelectron spectroscopy (UPS) to study whether or not these IFs changed the work function (WF) of the ETL. We suspected that the various functional groups (C_3_NH_2_, C_3_NH_3_I, and C_3_DMA) of the BPTI derivatives might have induced interfacial dipoles that could alter the WFs of the electrodes, thereby facilitating energy level alignment and leading to efficient carrier extraction [42,43,44,45]. We calculated the WFs of the ITO/ZnO, ITO/ZnO/CF_BPTI-C_3_NH_2_, ITO/ZnO/MeOH_BPTI-C_3_NH_3_I, and ITO/ZnO/CF_BPTI-C_3_DMA samples from the cutoff and valence band regions of the UPS spectra in Figure 4a. A change in surface status would affect the energy level alignment at the interface between the ETLs and the active layer. Here, we believed that the N and O atoms in BPTI-C_3_NH_2_, BPTI-C_3_NH_3_I, and BPTI-C_3_DMA would serve as hydrogen bond donors that could interact with the ZnO film to change its WF by forming net dipoles (from the molecular and surface dipoles) at the interface (Figure 4b) [46]. As listed in Table 3 The true WF of ITO/ZnO was −3.54 eV (consistent with the value reported in the literature); after embedding the BPTI-C_3_NH_2_, BPTI-C_3_NH_3_I, and BPTI-C_3_DMA interlayers, it decreased by 0.04, 0.05, and 0.01 eV, respectively, to give WFs of −3.50, −3.49, and −3.53 eV, respectively [47]. Thus, the WF of the ITO/ZnO substrate increased after embedding each of the BPTI-based IF layers [45]. Our findings suggest that the surface status (surface energy, morphology, WF) of the substrate altered the local BHJ morphology and changed the degree of electron extraction in the cathode. Because this morphology prevented unfavorable charge recombination at the interface with the ETL, the FFs and PCEs of the respective devices improved. Determining the possible effects of embedding IF layers (from variations in electronics properties to variations in morphologies) can be challenging; our approach will hopefully act as an example that will allow better understanding of the roles of interfaces within PSC devices.

To verify the effect of the interface layer on the ZnO film, we performed photoelectric flow measurements through a metal–semiconductor–metal (MSM) structure to determine the dipole direction and degree of charge accumulation. We used the device structure ITO/ZnO/IFL (CF_BPTI-C_3_NH_2_, MeOH_BPTI-C_3_NH_3_I, or CF_BPTI-C_3_DMA)/PC_71_BM/MoO_3_/Ag (here we applied PC_71_BM as the active layer, instead of PTB7-Th:PC_71_BM (Figure 5a)) to measure the photocurrent under an AM1.5 light source and check any deviation of the values of *V*_OC_ to determine whether changes occurred to their dipoles [48]. When we applied a negative bias to the device, the incoming electrons led to an accumulation of charge; when we applied a positive bias voltage, the internal electric fields would cancel each other, such that the value of *V*_OC_ would not be 0 V. Here we used the acceptor PC_71_BM as the active layer to observe the modifications of the ETL. If a dipole modification layer were present, more electrons would accumulate at the interface in this device state, with more positive bias being required to balance the internal electric field. The stronger the modification, the greater the deviation in the value of *V*_OC_. Therefore, by observing the changes in the values of *V*_OC_, we could judge whether the added IFL induced dipole modification [49]. Figure 5b reveals that the values of *V*_OC_ of the IFL-containing devices based on BPTI derivatives underwent significant deviations. Among them, the presence of BPTI-C_3_NH_3_I that had undergone SA with MeOH led to the largest deviation in the value of *V*_OC_; consistent with this finding, its PCE was also the best. In addition, we used transient photocurrent (TPC) measurements to investigate the charge extraction processes in devices prepared with and without an IFL (Appendix A) [14]. The charge extraction times of the control and BPTI-C_3_NH_3_I-modified devices were 0.671 and 0.651 µs, respectively, suggesting that the charge extraction efficiency of the device was promoted in the presence of BPTI-C_3_NH_3_I, resulting in a higher FF.

To test whether these BPTI derivatives could also improve the performance of other devices, we applied the BPTI-C_3_NH_3_I derivative in PM6:Y6:PC_71_BM ternary PSCs. Figure 6 and Table 2 present the results. The values of *V*_OC_, *J*_SC_, FF, and PCE of the device prepared without this IFL were 0.88 ± 0.01 V, 24.9 ± 0.39 mA cm^−2^, 71.8 ± 1.9%, and 15.6 ± 0.25%, respectively. When the BPTI-C_3_NH_3_I interface layer was present in the PM6:Y6:PC_71_BM ternary system, these values were 0.88 ± 0.01 V, 25.6 ± 0.70 mA cm^−2^, 73.3 ± 1.7%, and 16.5 ± 0.18%, respectively. Figure 6b presents the EQE spectra; the values of EQE–*J*_SC_ of the PSCs incorporating ZnO and the BPTI-C_3_NH_3_I-modified ZnO were 21.5 and 22.0 mA cm^−2^, respectively. Thus, this IFL could also improve the efficiency of a non-fullerene-based PSC.

## 3. Conclusions

We have applied materials presenting various functionalities—namely, BPTI-C_3_NH_2_, BPTI-C_3_NH_3_I, and BPTI-C_3_DMA—as IFLs in PSCs. The values of *J*_SC_ and FF of the devices were effectively modified after SA of their IFLs, thereby improving their PCEs. The average values of *J*_SC_, *V*_OC_, FF, and PCE for the PSC incorporating the MeOH_BPTI-C_3_NH_3_I–modified ZnO and PTB7-Th:PCBM active layer were 17.5 ± 0.84 mA cm^−2^, 0.82 ± 0.01 V, 70.6 ± 0.75%, and 10.2 ± 0.53%, respectively (cf. a best PCE for the control device of 9.1 ± 0.13%). This enhancement in performance resulted from improvements in the surface energy, energy level alignment, and carrier lifetimes. For PM6:Y6:PCBM-based ternary PSCs, the presence of the BPTI derivatives also resulted in efficient modification, with PCEs as high as 16.5 ± 0.18%, suggesting a universal effect for such BPTI derivatives as IFLs in PSC applications.

## Figures and Tables

**Figure 1 polymers-14-04466-f001:**
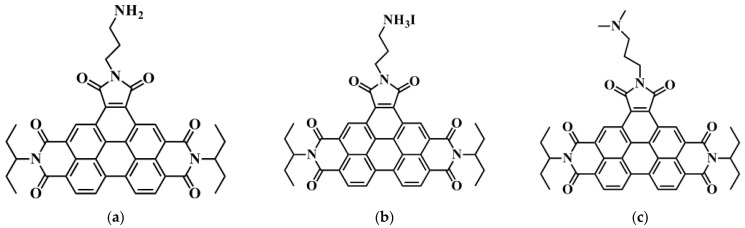
Chemical structures of the materials tested as interfacial layers. (**a**) BPTI-C_3_NH_2_; (**b**) BPTI-C_3_NH_3_I; (**c**) BPTI-C_3_DMA.

**Figure 2 polymers-14-04466-f002:**
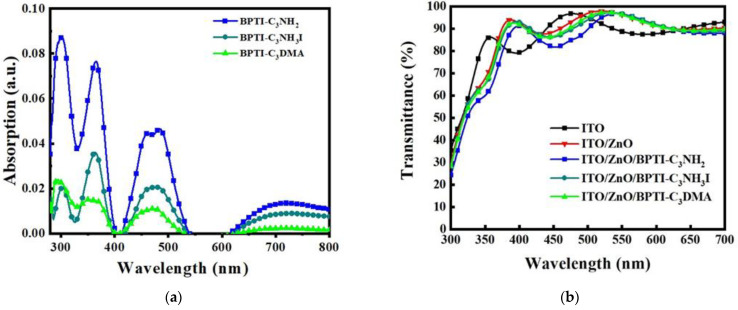
(**a**) UV–Vis absorption spectra of the BPTI films and (**b**) UV–Vis transmittance spectra of the ITO/ZnO/BPTI samples.

**Figure 3 polymers-14-04466-f003:**
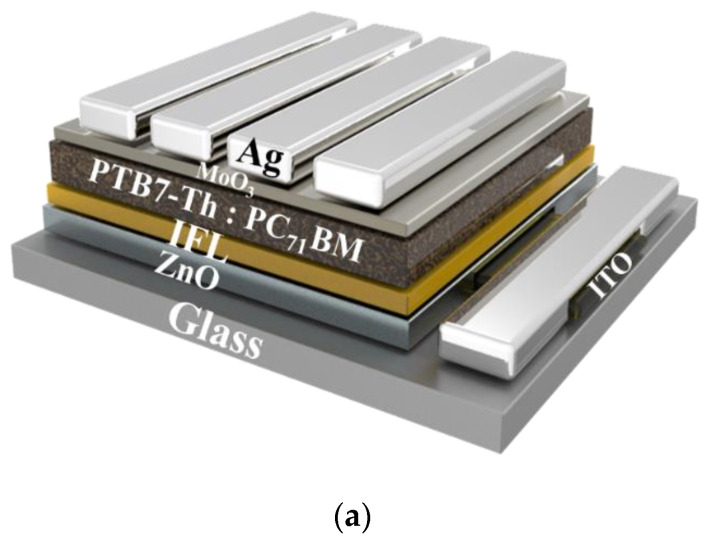
(**a**) Structure, (**b**) *J**–**V* curves, and (**c**) EQE spectra of the PSC devices.

**Figure 4 polymers-14-04466-f004:**
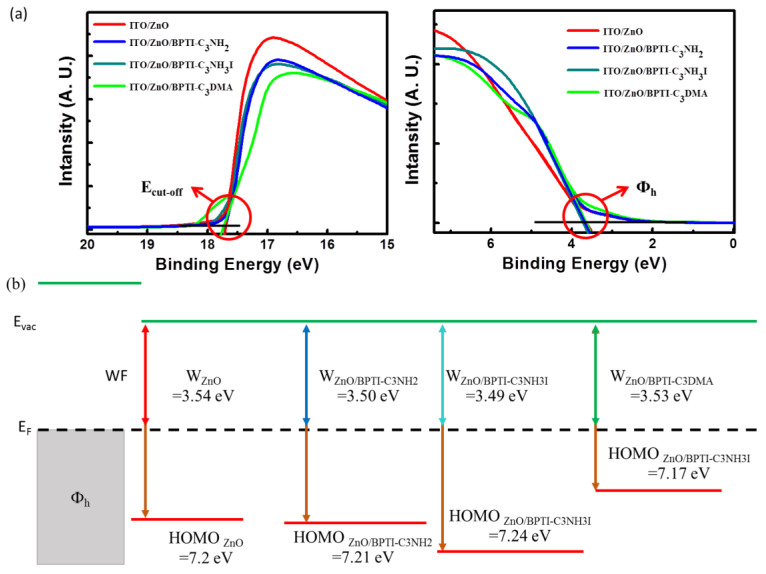
(**a**) UPS spectra of ZnO films prepared with and without IFLs and (**b**) energy levels of the materials.

**Figure 5 polymers-14-04466-f005:**
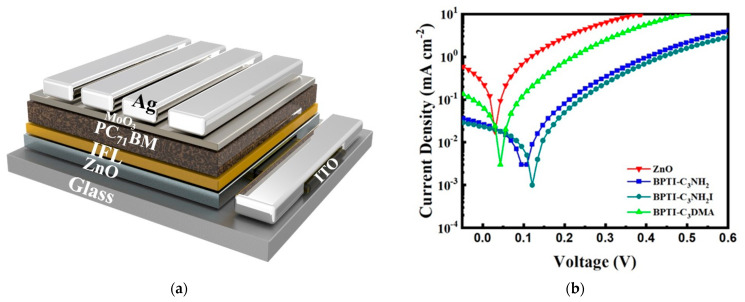
(**a**) MSM structure and (**b**) photoelectric flow curves of the devices.

**Figure 6 polymers-14-04466-f006:**
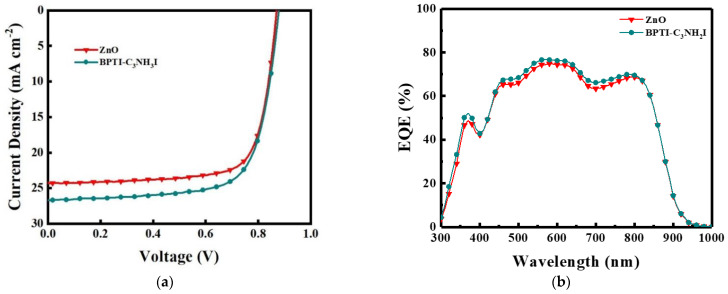
(**a**) *J*–*V* curves and (**b**) EQE spectra of PM6:Y6:PCBM−based PSCs.

**Table 1 polymers-14-04466-t001:** Contact angles and surface energies of the samples.

	*θ*_water_ (°)	*θ*_DIM_ (°)	*γ*_polar_ (mN m^−1^)	*γ*_dispersive_ (mN m^−1^)	*γ*_total_ (mN m^−1^)
ZnO ^a^	40.45	26.91	26.19	45.58	71.77
ZnO/BPTI-C_3_NH_2_ ^a^	54.65	24.67	19.11	46.37	65.48
ZnO/BPTI-C_3_NH_3_I ^a^	60.60	26.95	16.36	45.57	61.93
ZnO/BPTI-C_3_DMA ^a^	46.76	20.74	22.71	47.61	70.32
ZnO ^b^	-	-	20.87	45.45	66.32
ZnO/BPTI-C3NH_2_ ^b^	-	-	12.95	46.27	59.22
ZnO/BPTI-C_3_NH_3_I ^b^	-	-	10.19	45.43	55.62
ZnO/BPTI-C_3_DMA ^b^	-	-	16.65	47.56	64.21

^a^ Calculated using the Wu model. ^b^ Calculated using the OWRK model.

**Table 2 polymers-14-04466-t002:** *J*–*V* properties of the PSC devices.

Sample	Solvent Annealing (SA)	*J*_SC_ (mA cm^–2^)	*V*_OC_ (V)	FF (%)	PCE (%)	PCE_Best_ (%)
	Active layer:/PTB7-Th:PC71BM					
Without IFL	−	0.82 ± 0.01	17.3 ± 0.39	63.7 ± 2.0	9.1 ± 0.13	9.3
BPTI−C_3_NH_2_	Without	0.82 ± 0.01	16.81 ± 0.31	67.1 ± 1.20	9.3 ± 0.13	9.4
	CF	0.82 ± 0.01	17.1 ± 0.31	67.9 ± 3.47	9.6 ± 0.31	9.9
BPTI−C_3_NH_3_I	Without	0.81 ± 0.01	16.7 ± 0.18	68.4 ± 2.42	9.3 ± 0.48	9.4
	CF	0.82 ± 0.01	17.1 ± 0.31	70.5 ± 1.20	9.9 ± 0.11	10.0
	MeOH	0.82 ± 0.01	17.5 ± 0.84	70.6 ± 0.75	10.2 ± 0.53	10.8
BPTI−C_3_DMA	Without	0.81 ± 0.01	16.61 ± 0.31	68.1 ± 1.20	9.1 ± 0.11	9.2
	CF	0.81 ± 0.01	17.2 ± 0.47	67.4 ± 0.66	9.3 ± 0.26	9.6
	Active layer: PM6:Y6:PC_71_BM					
Without IFL	−	0.88 ± 0.01	24.9 ± 0.39	71.8 ± 1.9	15.6 ± 0.25	15.8
BPTI−C_3_NH_3_I	MeOH	0.88 ± 0.01	25.6 ± 0.70	73.3 ± 1.7	16.5 ± 0.18	16.8

**Table 3 polymers-14-04466-t003:** WFs of ZnO films prepared with and without IFLs, determined using UPS.

	*E*_Cut off_ (eV)	*Φ*_h_ (eV)	WF (eV)	HOMO (eV)
ZnO	17.68	3.66	3.54	7.20
ZnO/BPTI-C_3_NH_2_	17.72	3.71	3.50	7.21
ZnO/BPTI-C_3_NH_3_I	17.73	3.75	3.49	7.24
ZnO/BPTI-C_3_DMA	17.69	3.64	3.53	7.17

## Data Availability

The raw data presented in this study are available on request from the corresponding author.

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
