# Peer review of "Improving the Performance of Polymer Solar Cells with Benzo[ghi]perylenetriimide-Based Small-Molecules as Interfacial Layers"

_polymers, 2022, doi:10.3390/polym14204466_

Round 1
Reviewer 1 Report
please see the attachment

Author Response
Journal: Polymers (ISSN 2073-4360)
Manuscript ID: polymers-1943334
Type: Article
Title: Improving the Performance of Polymer Solar Cells with Benzo[ghi]perylenetriimide-Based Small Molecules as Interfacial Layers
Authors: Yang-Yen Yu,* Hung-Cheng Chen, Kai-Yu Shih, Yan-Cheng Peng, Bing-Huang Jiang, Chao-I Liu, Ming-Wei Hsu, Chi-Ching Kuo, Chih-Ping Chen*
Section: Polymer Applications
Special Issue: Applications of Polymers in Energy and Environmental Sciences II
Reviewer 1
- Line 84-101 the absorption/transmission spectra are not deeply analyzed; reporting the solution spectra vs film of the different BPTI could show evidence of the desired self-assembly. A similar comparison should be done with films deposed on quartz and on ITO/ZnO substrates.
Response: Thank you for all of your comments. We have added the following description to the revised manuscript:
Page 3: Figure S4 provides the UV–Vis absorption spectra of the BPTI derivatives as solutions in CHCl3 and in the form of glass/BPTI substrates. As indicated in Figure 2(a), the absorptions of the BPTI films were located mainly in the UV region and at wavelengths between 400 and 520 nm. The absorptions of the solid films were slightly red-shifted when compared with those in solution status (Figure S4a). The absorptions of the thin films were similar on the different substrates, suggesting that the nature of the substrate had only a minimal effect.
- (b)
Figure S4. (a) UV–Vis absorption spectra of the BPTI derivatives (a) as solutions in CHCl3 and (b) in the form of glass/BPTI substrates.
- Line 104-112 The rugosity measured by AFM does not look significantly different. The authors should report on how many samples and areas were measured to give a statistical relevance to their data.
Response: The similarity of the AFM images of the samples was due to the presence of the ultrathin (ca. 5 nm) interfacial layer. We suspect that the BPTIs passivated the defects of ZnO, resulting in the smooth surface. The AFM determined area of 5 μm ´ 5 μm could provide a macroscopic view of the samples. In general, we performed two sample measurements at various locations of the determined surface to obtain reliable AFM image data.
- The surface energy is calculated using the Wu model, however, it is not the best model for surface energy> 40mN/m. The OWRK would have been a more reliable model.
Response: We have recalculated the surface energy using the OWRK model and updated our results in Table 1. This model provided a similar relationship. We have added the following text to the revised manuscript:
Page 4: To double-check the data, we used the Owens–Wendt–Rabel–Kaelble (OWRK) model to calculate the surface energies of the various samples. Table 1 reveals that the results were similar, with the same trends in the changes in the surface energies of the respective samples.
Table 1. Contact angles and surface energies of the samples.
|
θwater (°) |
θDIM (°) |
γpolar (mN m–1) |
γdispersive (mN m–1) |
γtotal (mN m–1) |
ZnOa |
40.45 |
26.91 |
26.19 |
45.58 |
71.77 |
ZnO/BPTI-C3NH2a |
54.65 |
24.67 |
19.11 |
46.37 |
65.48 |
ZnO/BPTI-C3NH3Ia |
60.60 |
26.95 |
16.36 |
45.57 |
61.93 |
ZnO/BPTI-C3DMAa |
46.76 |
20.74 |
22.71 |
47.61 |
70.32 |
ZnOb |
– |
– |
20.87 |
45.45 |
66.32 |
ZnO/BPTI-C3NH2b |
– |
– |
12.95 |
46.27 |
59.22 |
ZnO/BPTI-C3NH3Ib |
– |
– |
10.19 |
45.43 |
55.62 |
ZnO/BPTI-C3DMAb |
– |
– |
16.65 |
47.56 |
64.21 |
a Calculated using the Wu model.
b Calculated using the OWRK model.
- The electrical characterization of the devices and the effect of the BPTI clearly appear. However the authors do not show any surface or morphological characterization of the interlayer, the vapour annealing in CF (and MeOH) appears to have an effect of the BPTI self-assembly.
Response: Because of the ultrathin nature of the BPTI layer, we would not be able to observe any differences in the AFM data after solvent annealing (CF or MeOH). We did, however, observed significant changes in the contact angles after modification. We have added the following description to our revised manuscript to address the Reviewer’s concerns.
Page 4: The orientation of a small molecule–based IFL can be adjusted through solvent vapor annealing (SA), with the optimized surface properties of the IFL directly affecting the performance of corresponding devices.[1] Because of the similar chemical structures of the three IFLs, we evaluated the effect only of MeOH (polar protic solvent) on BPTI-C3NH3I, which has high polarity due to its ammonium iodide functionality. Table S1 presents the contact angles and surface energies of the IFLs prepared with and without SA. We observed a higher value of θwater and a lower value of θDIM for the sample after SA with CF, suggesting that the ammonium iodide groups were embedded at the bottom of the film. After treatment with MeOH, the sample had a lower value of θwater with a higher value of θDIM, implying that the ammonium iodide units were distributed mainly on the surface of the IFL. These variations in surface orientation led to different surface energies for the CF- and MeOH-treated samples (60.73 and 62.91 mN m–1, respectively).
Table S1. Contact angles and surface energies of SA-treated IFLs.
ZnO/BPTI-C3NH3I |
θwater (°) |
θDIM (°) |
γpolar (mN m–1) |
γdispersive (mN m–1) |
γtotal (mN m–1) |
As-coated |
60.60 |
26.95 |
16.36 |
45.57 |
61.93 |
CF_SA |
66.34 |
21.49 |
13.34 |
47.38 |
60.73 |
MeOH_SA |
55.00 |
32.65 |
19.55 |
43.36 |
62.91 |
I find the article suitable for publication after major revisions, where the authors improved the discussion on the BPTI interlayer's morphological and interfacial effects.
Response: Thank you for all of your comments. Our modified manuscript is greatly improved.
References
- Ito, S.; Akiyama, H.; Sekizawa, R.; Mori, M.; Yoshida, M.; Kihara, H. Light-Induced Reworkable Adhesives Based on ABA-type Triblock Copolymers with Azopolymer Termini. ACS Applied Materials & Interfaces 2018, 10, 32649-32658, doi:10.1021/acsami.8b09319.
Journal: Polymers (ISSN 2073-4360)
Manuscript ID: polymers-1943334
Type: Article
Title: Improving the Performance of Polymer Solar Cells with Benzo[ghi]perylenetriimide-Based Small Molecules as Interfacial Layers
Authors: Yang-Yen Yu,* Hung-Cheng Chen, Kai-Yu Shih, Yan-Cheng Peng, Bing-Huang Jiang, Chao-I Liu, Ming-Wei Hsu, Chi-Ching Kuo, Chih-Ping Chen*
Section: Polymer Applications
Special Issue: Applications of Polymers in Energy and Environmental Sciences II
Reviewer 1
- Line 84-101 the absorption/transmission spectra are not deeply analyzed; reporting the solution spectra vs film of the different BPTI could show evidence of the desired self-assembly. A similar comparison should be done with films deposed on quartz and on ITO/ZnO substrates.
Response: Thank you for all of your comments. We have added the following description to the revised manuscript:
Page 3: Figure S4 provides the UV–Vis absorption spectra of the BPTI derivatives as solutions in CHCl3 and in the form of glass/BPTI substrates. As indicated in Figure 2(a), the absorptions of the BPTI films were located mainly in the UV region and at wavelengths between 400 and 520 nm. The absorptions of the solid films were slightly red-shifted when compared with those in solution status (Figure S4a). The absorptions of the thin films were similar on the different substrates, suggesting that the nature of the substrate had only a minimal effect.
- (b)
Figure S4. (a) UV–Vis absorption spectra of the BPTI derivatives (a) as solutions in CHCl3 and (b) in the form of glass/BPTI substrates.
- Line 104-112 The rugosity measured by AFM does not look significantly different. The authors should report on how many samples and areas were measured to give a statistical relevance to their data.
Response: The similarity of the AFM images of the samples was due to the presence of the ultrathin (ca. 5 nm) interfacial layer. We suspect that the BPTIs passivated the defects of ZnO, resulting in the smooth surface. The AFM determined area of 5 μm ´ 5 μm could provide a macroscopic view of the samples. In general, we performed two sample measurements at various locations of the determined surface to obtain reliable AFM image data.
- The surface energy is calculated using the Wu model, however, it is not the best model for surface energy> 40mN/m. The OWRK would have been a more reliable model.
Response: We have recalculated the surface energy using the OWRK model and updated our results in Table 1. This model provided a similar relationship. We have added the following text to the revised manuscript:
Page 4: To double-check the data, we used the Owens–Wendt–Rabel–Kaelble (OWRK) model to calculate the surface energies of the various samples. Table 1 reveals that the results were similar, with the same trends in the changes in the surface energies of the respective samples.
Table 1. Contact angles and surface energies of the samples.
|
θwater (°) |
θDIM (°) |
γpolar (mN m–1) |
γdispersive (mN m–1) |
γtotal (mN m–1) |
ZnOa |
40.45 |
26.91 |
26.19 |
45.58 |
71.77 |
ZnO/BPTI-C3NH2a |
54.65 |
24.67 |
19.11 |
46.37 |
65.48 |
ZnO/BPTI-C3NH3Ia |
60.60 |
26.95 |
16.36 |
45.57 |
61.93 |
ZnO/BPTI-C3DMAa |
46.76 |
20.74 |
22.71 |
47.61 |
70.32 |
ZnOb |
– |
– |
20.87 |
45.45 |
66.32 |
ZnO/BPTI-C3NH2b |
– |
– |
12.95 |
46.27 |
59.22 |
ZnO/BPTI-C3NH3Ib |
– |
– |
10.19 |
45.43 |
55.62 |
ZnO/BPTI-C3DMAb |
– |
– |
16.65 |
47.56 |
64.21 |
a Calculated using the Wu model.
b Calculated using the OWRK model.
- The electrical characterization of the devices and the effect of the BPTI clearly appear. However the authors do not show any surface or morphological characterization of the interlayer, the vapour annealing in CF (and MeOH) appears to have an effect of the BPTI self-assembly.
Response: Because of the ultrathin nature of the BPTI layer, we would not be able to observe any differences in the AFM data after solvent annealing (CF or MeOH). We did, however, observed significant changes in the contact angles after modification. We have added the following description to our revised manuscript to address the Reviewer’s concerns.
Page 4: The orientation of a small molecule–based IFL can be adjusted through solvent vapor annealing (SA), with the optimized surface properties of the IFL directly affecting the performance of corresponding devices.[1] Because of the similar chemical structures of the three IFLs, we evaluated the effect only of MeOH (polar protic solvent) on BPTI-C3NH3I, which has high polarity due to its ammonium iodide functionality. Table S1 presents the contact angles and surface energies of the IFLs prepared with and without SA. We observed a higher value of θwater and a lower value of θDIM for the sample after SA with CF, suggesting that the ammonium iodide groups were embedded at the bottom of the film. After treatment with MeOH, the sample had a lower value of θwater with a higher value of θDIM, implying that the ammonium iodide units were distributed mainly on the surface of the IFL. These variations in surface orientation led to different surface energies for the CF- and MeOH-treated samples (60.73 and 62.91 mN m–1, respectively).
Table S1. Contact angles and surface energies of SA-treated IFLs.
ZnO/BPTI-C3NH3I |
θwater (°) |
θDIM (°) |
γpolar (mN m–1) |
γdispersive (mN m–1) |
γtotal (mN m–1) |
As-coated |
60.60 |
26.95 |
16.36 |
45.57 |
61.93 |
CF_SA |
66.34 |
21.49 |
13.34 |
47.38 |
60.73 |
MeOH_SA |
55.00 |
32.65 |
19.55 |
43.36 |
62.91 |
I find the article suitable for publication after major revisions, where the authors improved the discussion on the BPTI interlayer's morphological and interfacial effects.
Response: Thank you for all of your comments. Our modified manuscript is greatly improved.
References
- Ito, S.; Akiyama, H.; Sekizawa, R.; Mori, M.; Yoshida, M.; Kihara, H. Light-Induced Reworkable Adhesives Based on ABA-type Triblock Copolymers with Azopolymer Termini. ACS Applied Materials & Interfaces 2018, 10, 32649-32658, doi:10.1021/acsami.8b09319.

Reviewer 2 Report
The authors reports about the application of three BPTI as ETL in OPV.
The authors recently published a similar work on pervskite PV (https://doi.org/10.1016/j.dyepig.2021.109385) and as a non-fulerene acceptor (https://doi.org/10.1002/chem.201804088).
- Line 84-101 the absorption/transmission spectra are not deeply analyzed; reporting the solution spectra vs film of the different BPTI could show evidence of the desired self-assembly. A similar comparison should be done with films deposed on quartz and on ITO/ZnO substrates.
- Line 104-112 The rugosity measured by AFM does not look significantly different. The authors should report on how many samples and areas were measured to give a statistical relevance to their data.
- The surface energy is calculated using the Wu model, however, it is not the best model for surface energy> 40mN/m. The OWRK would have been a more reliable model.
- The electrical characterization of the devices and the effect of the BPTI clearly appear. However the authors do not show any surface or morpological characterization of the the interlayer, the vapour annealing in CF (and MeOH) appears to have an effect of the BPTI self-assembly.
I find the article suitable for publication after major revisions, where the authors improved the discussion on the BPTI interlayer's morphological and interfacial effects.
Author Response
Journal: Polymers (ISSN 2073-4360)
Manuscript ID: polymers-1943334
Type: Article
Title: Improving the Performance of Polymer Solar Cells with Benzo[ghi]perylenetriimide-Based Small Molecules as Interfacial Layers
Authors: Yang-Yen Yu,* Hung-Cheng Chen, Kai-Yu Shih, Yan-Cheng Peng, Bing-Huang Jiang, Chao-I Liu, Ming-Wei Hsu, Chi-Ching Kuo, Chih-Ping Chen*
Section: Polymer Applications
Special Issue: Applications of Polymers in Energy and Environmental Sciences II
Reviewer #2
Although authors did study the structure-property-relationships of the interracial materials
in the OPV, there has been issues that should be addressed in the manuscript before the
publication.
- It has been reported that, difference In the surface energy between the donor and acceptor materials plays an important role in deciding the miscibility in the active layer [Mater. Horiz. 2021, 8, 1008−1016; ACS Appl. Polym. Mater. 2021, 3, 5, 2759–2767; ACS Appl. Polym. Mater. 2021, 3, 1923−1931.]. Like these reports, can authors try to establish a correlation between the surface energy of the interracial materials or difference in the surface energy of the interfacial layer with active layer surface energy with PCE of solar cell?
Response: Thank you for all of your comments. The performance of the OPV was determined by various factors, so it would be difficult to reach a conclusion simply by considering only one or two of them. To address the Reviewer’s concerns, we have added the following discussion to the revised manuscript regarding the correlation between the surface energy of the interfacial materials and the active layer:
Page 4: The miscibility of donor and acceptor moieties, a characteristic that can be evaluated from the surface energy, is a major factor that can affect the blend morphology.[2-4] Previous reports have indicated that the surface energy of a substrate can alter the phase separation morphology of the active layer,[5] with the driving force possibly being a large difference in surface energy (or miscibility) between the two components.[6] Thus, we examined the effect of changes in the surface energies of the substrates upon the variations in their blend film morphologies.
- Figure 3a should be corrected make it clear (Labels).
Response: We have corrected this figure according in the revised manuscript.
Figure 3. (a) Structure of the PSC devices.
- EQE should be included for the Y6 based OPVs.
- Integrated Jsc should be included in the manuscript.
Responses: In the previous version of the data, we observed a large mismatch between the values of J-V-Jsc and EQE-Jsc. We have re-examined the data and provide the following description in our revised manuscript:
Page 10: From the EQE spectra and the solar flux, we calculated the values of EQE–JSC of the PSCs incorporating ZnO and the BPTI-C3NH3I–modified ZnO to be 15.2 and 15.6 mA cm–2, respectively. The mismatch arose from various factors, including the measurement conditions at the solar simulator not being the same as those during the EQE measurements.[7]
Figure 3. (c) EQE spectra of the PSC devices.
Page 8: Figure 6b presents the EQE spectra; the values of EQE–JSC of the PSCs incorporating ZnO and the BPTI-C3NH3I–modified ZnO were 21.5 and 22.0 mA cm–2, respectively.
Figure 6. (b) EQE spectra of PM6:Y6:PCBM-based PSCs.
- Surface morphology of the new interfacial layers should be included in the main
manuscript.
Response: Please see our reply to Q4 from Reviewer 1.
- Charge transport studies should be performed with and without the interfacial material in the device to understand the role of interfacial material in the charge transport
Response: We have added the following description to our revised manuscript:
Page 8: In addition, we used transient photocurrent (TPC) measurements to investigate the charge extraction processes in devices prepared with and without an IFL (Figure S7).[8] The charge extraction times of the control and BPTI-C3NH3I–modified devices were 0.671 and 0.651 µs, respectively, suggesting that the charge extraction efficiency of the device was promoted in the presence of BPTI-C3NH3I, resulting in a higher FF.
Figure S7. Normalized TPC data for the control and BPTI-C3NH3I–modified devices.
References
- Ito, S.; Akiyama, H.; Sekizawa, R.; Mori, M.; Yoshida, M.; Kihara, H. Light-Induced Reworkable Adhesives Based on ABA-type Triblock Copolymers with Azopolymer Termini. ACS Applied Materials & Interfaces 2018, 10, 32649-32658, doi:10.1021/acsami.8b09319.
- Kranthiraja, K.; Saeki, A. Impact of Sequential Fluorination of Donor and/or Acceptor Polymers on the Efficiency and Morphology of All-Polymer Solar Cells. ACS Applied Polymer Materials 2021, 3, 2759-2767, doi:10.1021/acsapm.1c00288.
- Chen, D.; Liu, S.; Liu, J.; Han, J.; Chen, L.; Chen, Y. Regulation of the Miscibility of the Active Layer by Random Terpolymer Acceptors to Realize High-Performance All-Polymer Solar Cells. ACS Applied Polymer Materials 2021, 3, 1923-1931, doi:10.1021/acsapm.1c00004.
- Du, F.; Wang, H.; Zhang, Z.; Yang, L.; Cao, J.; Yu, J.; Tang, W. An unfused-ring acceptor with high side-chain economy enabling 11.17% as-cast organic solar cells. Materials Horizons 2021, 8, 1008-1016, doi:10.1039/D0MH01585G.
- Bulliard, X.; Ihn, S.-G.; Yun, S.; Kim, Y.; Choi, D.; Choi, J.-Y.; Kim, M.; Sim, M.; Park, J.-H.; Choi, W.; et al. Enhanced Performance in Polymer Solar Cells by Surface Energy Control. Advanced Functional Materials 2010, 20, 4381-4387, doi:https://doi.org/10.1002/adfm.201000960.
- Jiang, B.H.; Peng, Y.-J.; Huang, Y.-C.; Jeng, R.-J.; Shieh, T.-S.; Huang, C.-I.; Chen, C.-P. Greater miscibility and energy level alignment of conjugated polymers enhance the optoelectronic properties of ternary blend films in organic photovoltaics. Dyes and Pigments 2021, 193, 109543, doi:https://doi.org/10.1016/j.dyepig.2021.109543.
- Saliba, M.; Etgar, L. Current Density Mismatch in Perovskite Solar Cells. ACS Energy Letters 2020, 5, 2886-2888, doi:10.1021/acsenergylett.0c01642.
- Sun, R.; Wu, Y.; Yang, X.; Gao, Y.; Chen, Z.; Li, K.; Qiao, J.; Wang, T.; Guo, J.; Liu, C.; et al. Single-Junction Organic Solar Cells with 19.17% Efficiency Enabled by Introducing One Asymmetric Guest Acceptor. Advanced Materials 2022, 34, 2110147, doi:https://doi.org/10.1002/adma.202110147.

Round 2
Reviewer 2 Report
The authors replied only to reviewer#2, ignoring all my comments in their answers.
I suggest the publication of the article after major revisions as detailed in my first report
Author Response
Journal: Polymers (ISSN 2073-4360)
Manuscript ID: polymers-1943334
Type: Article
Title: Improving the Performance of Polymer Solar Cells with Benzo[ghi]perylenetriimide-Based Small Molecules as Interfacial Layers
Authors: Yang-Yen Yu,* Hung-Cheng Chen, Kai-Yu Shih, Yan-Cheng Peng, Bing-Huang Jiang, Chao-I Liu, Ming-Wei Hsu, Chi-Ching Kuo, Chih-Ping Chen*
Section: Polymer Applications
Special Issue: Applications of Polymers in Energy and Environmental Sciences II
Dear reviewer, we are sincerely sorry for uploading our reply for you to the wrong place and the inconvenience we made.
Reviewer 1
- Line 84-101 the absorption/transmission spectra are not deeply analyzed; reporting the solution spectra vs film of the different BPTI could show evidence of the desired self-assembly. A similar comparison should be done with films deposed on quartz and on ITO/ZnO substrates.
Response: Thank you for all of your comments. We have added the following description to the revised manuscript:
Page 3: Figure S4 provides the UV–Vis absorption spectra of the BPTI derivatives as solutions in CHCl3 and in the form of glass/BPTI substrates. As indicated in Figure 2(a), the absorptions of the BPTI films were located mainly in the UV region and at wavelengths between 400 and 520 nm. The absorptions of the solid films were slightly red-shifted when compared with those in solution status (Figure S4a). The absorptions of the thin films were similar on the different substrates, suggesting that the nature of the substrate had only a minimal effect.
- (b)
Figure S4. (a) UV–Vis absorption spectra of the BPTI derivatives (a) as solutions in CHCl3 and (b) in the form of glass/BPTI substrates.
- Line 104-112 The rugosity measured by AFM does not look significantly different. The authors should report on how many samples and areas were measured to give a statistical relevance to their data.
Response: The similarity of the AFM images of the samples was due to the presence of the ultrathin (ca. 5 nm) interfacial layer. We suspect that the BPTIs passivated the defects of ZnO, resulting in the smooth surface. The AFM determined area of 5 μm ´ 5 μm could provide a macroscopic view of the samples. In general, we performed two sample measurements at various locations of the determined surface to obtain reliable AFM image data.
- The surface energy is calculated using the Wu model, however, it is not the best model for surface energy> 40mN/m. The OWRK would have been a more reliable model.
Response: We have recalculated the surface energy using the OWRK model and updated our results in Table 1. This model provided a similar relationship. We have added the following text to the revised manuscript:
Page 4: To double-check the data, we used the Owens–Wendt–Rabel–Kaelble (OWRK) model to calculate the surface energies of the various samples. Table 1 reveals that the results were similar, with the same trends in the changes in the surface energies of the respective samples.
Table 1. Contact angles and surface energies of the samples.
|
θwater (°) |
θDIM (°) |
γpolar (mN m–1) |
γdispersive (mN m–1) |
γtotal (mN m–1) |
ZnOa |
40.45 |
26.91 |
26.19 |
45.58 |
71.77 |
ZnO/BPTI-C3NH2a |
54.65 |
24.67 |
19.11 |
46.37 |
65.48 |
ZnO/BPTI-C3NH3Ia |
60.60 |
26.95 |
16.36 |
45.57 |
61.93 |
ZnO/BPTI-C3DMAa |
46.76 |
20.74 |
22.71 |
47.61 |
70.32 |
ZnOb |
– |
– |
20.87 |
45.45 |
66.32 |
ZnO/BPTI-C3NH2b |
– |
– |
12.95 |
46.27 |
59.22 |
ZnO/BPTI-C3NH3Ib |
– |
– |
10.19 |
45.43 |
55.62 |
ZnO/BPTI-C3DMAb |
– |
– |
16.65 |
47.56 |
64.21 |
a Calculated using the Wu model.
b Calculated using the OWRK model.
- The electrical characterization of the devices and the effect of the BPTI clearly appear. However the authors do not show any surface or morphological characterization of the interlayer, the vapour annealing in CF (and MeOH) appears to have an effect of the BPTI self-assembly.
Response: Because of the ultrathin nature of the BPTI layer, we would not be able to observe any differences in the AFM data after solvent annealing (CF or MeOH). We did, however, observed significant changes in the contact angles after modification. We have added the following description to our revised manuscript to address the Reviewer’s concerns.
Page 4: The orientation of a small molecule–based IFL can be adjusted through solvent vapor annealing (SA), with the optimized surface properties of the IFL directly affecting the performance of corresponding devices.[1] Because of the similar chemical structures of the three IFLs, we evaluated the effect only of MeOH (polar protic solvent) on BPTI-C3NH3I, which has high polarity due to its ammonium iodide functionality. Table S1 presents the contact angles and surface energies of the IFLs prepared with and without SA. We observed a higher value of θwater and a lower value of θDIM for the sample after SA with CF, suggesting that the ammonium iodide groups were embedded at the bottom of the film. After treatment with MeOH, the sample had a lower value of θwater with a higher value of θDIM, implying that the ammonium iodide units were distributed mainly on the surface of the IFL. These variations in surface orientation led to different surface energies for the CF- and MeOH-treated samples (60.73 and 62.91 mN m–1, respectively).
Table S1. Contact angles and surface energies of SA-treated IFLs.
ZnO/BPTI-C3NH3I |
θwater (°) |
θDIM (°) |
γpolar (mN m–1) |
γdispersive (mN m–1) |
γtotal (mN m–1) |
As-coated |
60.60 |
26.95 |
16.36 |
45.57 |
61.93 |
CF_SA |
66.34 |
21.49 |
13.34 |
47.38 |
60.73 |
MeOH_SA |
55.00 |
32.65 |
19.55 |
43.36 |
62.91 |
I find the article suitable for publication after major revisions, where the authors improved the discussion on the BPTI interlayer's morphological and interfacial effects.
Response: Thank you for all of your comments. Our modified manuscript is greatly improved.
References
- Ito, S.; Akiyama, H.; Sekizawa, R.; Mori, M.; Yoshida, M.; Kihara, H. Light-Induced Reworkable Adhesives Based on ABA-type Triblock Copolymers with Azopolymer Termini. ACS Applied Materials & Interfaces 2018, 10, 32649-32658, doi:10.1021/acsami.8b09319.
Journal: Polymers (ISSN 2073-4360)
Manuscript ID: polymers-1943334
Type: Article
Title: Improving the Performance of Polymer Solar Cells with Benzo[ghi]perylenetriimide-Based Small Molecules as Interfacial Layers
Authors: Yang-Yen Yu,* Hung-Cheng Chen, Kai-Yu Shih, Yan-Cheng Peng, Bing-Huang Jiang, Chao-I Liu, Ming-Wei Hsu, Chi-Ching Kuo, Chih-Ping Chen*
Section: Polymer Applications
Special Issue: Applications of Polymers in Energy and Environmental Sciences II
Reviewer 1
- Line 84-101 the absorption/transmission spectra are not deeply analyzed; reporting the solution spectra vs film of the different BPTI could show evidence of the desired self-assembly. A similar comparison should be done with films deposed on quartz and on ITO/ZnO substrates.
Response: Thank you for all of your comments. We have added the following description to the revised manuscript:
Page 3: Figure S4 provides the UV–Vis absorption spectra of the BPTI derivatives as solutions in CHCl3 and in the form of glass/BPTI substrates. As indicated in Figure 2(a), the absorptions of the BPTI films were located mainly in the UV region and at wavelengths between 400 and 520 nm. The absorptions of the solid films were slightly red-shifted when compared with those in solution status (Figure S4a). The absorptions of the thin films were similar on the different substrates, suggesting that the nature of the substrate had only a minimal effect.
- (b)
Figure S4. (a) UV–Vis absorption spectra of the BPTI derivatives (a) as solutions in CHCl3 and (b) in the form of glass/BPTI substrates.
- Line 104-112 The rugosity measured by AFM does not look significantly different. The authors should report on how many samples and areas were measured to give a statistical relevance to their data.
Response: The similarity of the AFM images of the samples was due to the presence of the ultrathin (ca. 5 nm) interfacial layer. We suspect that the BPTIs passivated the defects of ZnO, resulting in the smooth surface. The AFM determined area of 5 μm ´ 5 μm could provide a macroscopic view of the samples. In general, we performed two sample measurements at various locations of the determined surface to obtain reliable AFM image data.
- The surface energy is calculated using the Wu model, however, it is not the best model for surface energy> 40mN/m. The OWRK would have been a more reliable model.
Response: We have recalculated the surface energy using the OWRK model and updated our results in Table 1. This model provided a similar relationship. We have added the following text to the revised manuscript:
Page 4: To double-check the data, we used the Owens–Wendt–Rabel–Kaelble (OWRK) model to calculate the surface energies of the various samples. Table 1 reveals that the results were similar, with the same trends in the changes in the surface energies of the respective samples.
Table 1. Contact angles and surface energies of the samples.
|
θwater (°) |
θDIM (°) |
γpolar (mN m–1) |
γdispersive (mN m–1) |
γtotal (mN m–1) |
ZnOa |
40.45 |
26.91 |
26.19 |
45.58 |
71.77 |
ZnO/BPTI-C3NH2a |
54.65 |
24.67 |
19.11 |
46.37 |
65.48 |
ZnO/BPTI-C3NH3Ia |
60.60 |
26.95 |
16.36 |
45.57 |
61.93 |
ZnO/BPTI-C3DMAa |
46.76 |
20.74 |
22.71 |
47.61 |
70.32 |
ZnOb |
– |
– |
20.87 |
45.45 |
66.32 |
ZnO/BPTI-C3NH2b |
– |
– |
12.95 |
46.27 |
59.22 |
ZnO/BPTI-C3NH3Ib |
– |
– |
10.19 |
45.43 |
55.62 |
ZnO/BPTI-C3DMAb |
– |
– |
16.65 |
47.56 |
64.21 |
a Calculated using the Wu model.
b Calculated using the OWRK model.
- The electrical characterization of the devices and the effect of the BPTI clearly appear. However the authors do not show any surface or morphological characterization of the interlayer, the vapour annealing in CF (and MeOH) appears to have an effect of the BPTI self-assembly.
Response: Because of the ultrathin nature of the BPTI layer, we would not be able to observe any differences in the AFM data after solvent annealing (CF or MeOH). We did, however, observed significant changes in the contact angles after modification. We have added the following description to our revised manuscript to address the Reviewer’s concerns.
Page 4: The orientation of a small molecule–based IFL can be adjusted through solvent vapor annealing (SA), with the optimized surface properties of the IFL directly affecting the performance of corresponding devices.[1] Because of the similar chemical structures of the three IFLs, we evaluated the effect only of MeOH (polar protic solvent) on BPTI-C3NH3I, which has high polarity due to its ammonium iodide functionality. Table S1 presents the contact angles and surface energies of the IFLs prepared with and without SA. We observed a higher value of θwater and a lower value of θDIM for the sample after SA with CF, suggesting that the ammonium iodide groups were embedded at the bottom of the film. After treatment with MeOH, the sample had a lower value of θwater with a higher value of θDIM, implying that the ammonium iodide units were distributed mainly on the surface of the IFL. These variations in surface orientation led to different surface energies for the CF- and MeOH-treated samples (60.73 and 62.91 mN m–1, respectively).
Table S1. Contact angles and surface energies of SA-treated IFLs.
ZnO/BPTI-C3NH3I |
θwater (°) |
θDIM (°) |
γpolar (mN m–1) |
γdispersive (mN m–1) |
γtotal (mN m–1) |
As-coated |
60.60 |
26.95 |
16.36 |
45.57 |
61.93 |
CF_SA |
66.34 |
21.49 |
13.34 |
47.38 |
60.73 |
MeOH_SA |
55.00 |
32.65 |
19.55 |
43.36 |
62.91 |
I find the article suitable for publication after major revisions, where the authors improved the discussion on the BPTI interlayer's morphological and interfacial effects.
Response: Thank you for all of your comments. Our modified manuscript is greatly improved.
References
- Ito, S.; Akiyama, H.; Sekizawa, R.; Mori, M.; Yoshida, M.; Kihara, H. Light-Induced Reworkable Adhesives Based on ABA-type Triblock Copolymers with Azopolymer Termini. ACS Applied Materials & Interfaces 2018, 10, 32649-32658, doi:10.1021/acsami.8b09319.

Round 3
Reviewer 2 Report
I thank the authors for this new version and their answers. They convincingly answered to my comments and provided new experimental data to strengthen their findings.
I find this article suitable for publication in present form.